# Transcranial Photosensitizer-Free Laser Treatment of Glioblastoma in Rat Brain

**DOI:** 10.3390/ijms241813696

**Published:** 2023-09-05

**Authors:** Oxana Semyachkina-Glushkovskaya, Sergey Sokolovski, Ivan Fedosov, Alexander Shirokov, Nikita Navolokin, Alla Bucharskaya, Inna Blokhina, Andrey Terskov, Alexander Dubrovski, Valeria Telnova, Anna Tzven, Maria Tzoy, Arina Evsukova, Daria Zhlatogosrkaya, Viktoria Adushkina, Alexander Dmitrenko, Maria Manzhaeva, Valeria Krupnova, Alessio Noghero, Denis Bragin, Olga Bragina, Ekaterina Borisova, Jürgen Kurths, Edik Rafailov

**Affiliations:** 1Physics Department, Humboldt University, Newtonstrasse 15, 12489 Berlin, Germany; juergen.kurths@pik-potsdam.de; 2Department of Biology, Saratov State University, Astrakhanskaya Str. 83, 410012 Saratov, Russia; shirokov_a@ibppm.ru (A.S.); nik-navolokin@yandex.ru (N.N.); inna-474@yandex.ru (I.B.); terskow.andrey@gmail.com (A.T.); ler.vinnick2012@yandex.ru (V.T.); anna.kuzmina.270599@mail.ru (A.T.); arina-evsyukova@mail.ru (A.E.); eloveda@mail.ru (D.Z.); adushkina.info@mail.ru (V.A.); admitrenko2001@mail.ru (A.D.); mariamang1412@gmail.com (M.M.); krupnova_0110@mail.ru (V.K.); 3Optoelectronics and Biomedical Photonics Group, AIPT, Aston University, Birmingham B4 7ET, UK; e.rafailov@aston.ac.uk; 4Physics Department, Saratov State University, Astrakhanskaya Str. 83, 410012 Saratov, Russia; fedosov_optics@mail.ru (I.F.); paskalkamal@mail.ru (A.D.); dethaos@bk.ru (M.T.); 5Institute of Biochemistry and Physiology of Plants and Microorganisms, Russian Academy of Sciences, Prospekt Entuziastov 13, 410049 Saratov, Russia; 6Department of Pathological Anatomy, Saratov Medical State University, Bolshaya Kazachaya Str. 112, 410012 Saratov, Russia; allaalla_72@mail.ru; 7Lovelace Biomedical Research Institute, Albuquerque, NM 87108, USA; noghero@gmx.com (A.N.); dbragin@salud.unm.edu (D.B.); obragina@gmx.com (O.B.); 8Department of Neurology, School of Medicine, University of New Mexico, Albuquerque, NM 87131, USA; 9Institute of Electronics, Bulgarian Academy of Sciences, Tsarigradsko Chaussee Blvd. 72, 1784 Sofia, Bulgaria; ekaterina.borisova@gmail.com; 10Potsdam Institute for Climate Impact Research, Telegrafenberg A31, 14473 Potsdam, Germany; 11Centre for Analysis of Complex Systems, Sechenov First Moscow State Medical University Moscow, 119991 Moscow, Russia

**Keywords:** glioblastoma, phototherapy, quantum-dot 1267 nm laser, NIR laser oxidative cell damaging, lymphatic system

## Abstract

Over sixty years, laser technologies have undergone a technological revolution and become one of the main tools in biomedicine, particularly in neuroscience, neurodegenerative diseases and brain tumors. Glioblastoma is the most lethal form of brain cancer, with very limited treatment options and a poor prognosis. In this study on rats, we demonstrate that glioblastoma (GBM) growth can be suppressed by photosensitizer-free laser treatment (PS-free-LT) using a quantum-dot-based 1267 nm laser diode. This wavelength, highly absorbed by oxygen, is capable of turning triplet oxygen to singlet form. Applying 1267 nm laser irradiation for a 4 week course with a total dose of 12.7 kJ/cm^2^ firmly suppresses GBM growth and increases survival rate from 34% to 64%, presumably via LT-activated apoptosis, inhibition of the proliferation of tumor cells, a reduction in intracranial pressure and stimulation of the lymphatic drainage and clearing functions. PS-free-LT is a promising breakthrough technology in non- or minimally invasive therapy for superficial GBMs in infants as well as in adult patients with high photosensitivity or an allergic reaction to PSs.

## 1. Introduction

Progress in laser development has revolutionized research approaches in biomedicine, creating new fields in medical diagnostics and treatment, e.g., different types of linear and non-linear microscopy and spectroscopy, neurophotonics [1], phototherapy of neurodegenerative diseases [2], cancer photodynamic therapy (PDT) [3], and explicit cure of brain tumors.

Glioblastomas (GBMs) are the most common central nervous system (CNS) tumor encountered in adults, but they only represent approximately 8–12% of all pediatric CNS tumors [4,5,6]. Historically, pediatric GBMs were assumed to be similar to adult brain tumors, since they appear to be histologically identical. However, molecular, genetic, and biological data reveal that they demonstrate significant differences [5]. The only similarity is that pediatric GBMs are as aggressive and malignant as adult ones, with only a few patients achieving long-term survival despite a variety of therapies being applied [5]. 

Surgical resection is the first step in treating all patients with GBM [5,7,8]. Even if GBM resection is successfully performed, 80% of pediatric and adult GBMs recur within a 2 cm periphery of the initial resection site [9,10,11,12]. This is due to the infiltrative nature of these lesions, making it virtually impossible to achieve GBM with clear surgical margins without risking significant morbidity. Treating GBMs in infants with radiotherapy (RT) has demonstrated limitations, and late treatments have fewer positive outcomes than earlier ones [13]. Indeed, RT is the mainstay approach for children older than 3 years, because infants are more susceptible to the negative deleterious effects associated with radiation [14,15,16,17]. However, RT techniques such as hyper- and hypo-fractionations have not consistently been proven to be statistically beneficial for children with GBM [18]. At the same time, this treatment in infants can result in severe impairment of brain development, growth and cognitive abilities [14] due to direct ionization of DNA, causing not only DNA degradation in GBM cell nuclei but also DNA mutations in healthy neuronal cells surrounding the GBM [19]. Since laser treatment (LT) can only generate reactive oxygen species (ROS) and is unable to directly mutate nuclei DNA [20], this can make LT more favorable for therapy in pediatric GBM compared with the RT approach.

Another approach in treating children with GBM is the use of chemotherapy [5]. However, chemotherapeutical methods are still considered controversial and are not universally accepted [21,22], because chemotherapy if applied has intrinsic toxicity to the whole organism. Since LT is applied on the tumor area only, organismal toxicity can be negligibly low.

The extremely poor life expectancy for patients with GBM and the limitations of pediatric tumor therapy have made oncologists seek safer and more selective therapies, in particular in the field of photonics. This is why PDT was discussed and then invented as a promising least-invasive tool for the treatment of both pediatric and adult GBM [23,24,25,26,27,28,29,30,31]. PDT is a form of therapy that combines a light source and photosensitizers (PSs). Light activation of PSs transfers energy to molecular oxygen, generating singlet oxygen (^1^O_2_), which is a highly reactive and toxic species that rapidly causes oxidative damage, ultimately leading to cell death. Starting from 1980, Perria et. al. used PDT to significantly improve the life expectance of eight patients with GBM [30]. It took almost 30 years to increase the survival rate of patients with GBM to 1.5 years. The most successful results were shown by Cramer et al., who reported overall survival rates in those with newly diagnosed and recurrent GBMs of 28.0 and 40.0% at 2 years and 22.0 and 34.0% at 5 years, respectively [31]. 

Cancer alleviation is not only the result of direct tumor-cell PDT toxicity but also of a significant immunotherapeutic effect on the tumor [32,33,34,35]. The first evidence for anti-tumor immunity induced by PDT was demonstrated in mice [36]. The animals were able to resist a re-challenge with tumor cells after PDT. Immunogenic cancer cell death induced by PDT starts with the release of specific molecular patterns associated with tumor damage and generates anti-tumor immunity based on surface calreticulin, heat shock protein 70, secreted adenosine triphosphate, and high-mobility group box 1 protein [37,38,39,40,41]. The PDT-induced immune response causes the maturation of antigen-presenting cells, which present the tumor antigens to a specific subset of CDs+ T cells [42]. 

Immune system response against brain tumors is thought to be limited by the lack of a cerebral lymphatic drainage system. However, recently, the meningeal lymphatic vessels (MLVs) were re-discovered as being able to amplify the immune response against brain tumors [43,44]. The drainage and clearing functions of MLVs have been shown in our [45,46,47,48,49] and other studies [50,51] to be an important mechanism in keeping the health of the CNS. In our pilot experimental work, we have reported photostimulation of MLV functions [45,47,48,49,52,53] using a quantum-dot 1267 nm laser, which can stimulate the direct generation of singlet oxygen (^1^O_2_) from the triplet oxygen state without photosensitizers [54]. The results from in vitro studies clearly demonstrate the suppression of cancer cell progression with 1267 nm laser irradiation through the induction of apoptosis in different cancer cell lines [54,55,56,57]. The benefit of using this wavelength for in vivo tumor LT is its high optical transmission window for scalp and skull [58,59]. These studies clearly demonstrate that 1267 nm irradiation is capable of generating cellular oxidative stress in magnitudes that can trigger apoptotic events in cancer cells, bringing them to the brink of cell death. Later, it was shown that effectiveness of 1267 nm laser irradiation is directly allied with the highly localized surface plasmon polariton amplification effect generated on mitochondria crista [55,56]. These intriguing findings bring new strategies to overcome the limitations of GBM therapy in infants using innovative PS-free PDT, as well as in adult patients who have allergic responses to PSs. 

In this in vivo and ex vivo research on rats as well as on C6 glioma cells in in vitro experiments, we intend to demonstrate that the course of PS-free transcranial non-invasive LT is able to significantly suppress GBM growth in rat brain and positively affect survival rate via LT-induced oxidative stress causing cancer cell apoptosis, a decrease in GBM cell proliferation, and a reduction in intracranial pressure (ICP) mediated via LT stimulation of the lymphatic drainage and clearing functions.

## 2. Results

### 2.1. The Effects of an LT Course on GBM Progression and Survival Rate

It is well known that irradiation in the wavelength range from 1100 nm and longer begins to be absorbed by water and can cause significant heating effects on the water content in biological objects. It was demonstrated that brain functions are so sensitive to changes in temperature that an increase of 0.5 °C can cause significant alterations in cellular processes [60,61]. An increase of 1 °C or above can cause profound effects on neural network functioning. Therefore, we measured the temperature on the skull (scalp removed) and the top of the cortex using three laser powers of 70, 100 and 130 mW in rats with 4-week-old GBMs, when the tumor had reached the surface of the brain in all animals (Appendix A). Figure 1a and Appendix A illustrate the locations of 4-week-old GBMs on the brain surface. The temperature was analyzed directly on the surface of the GBM and on the brain tissues surrounding the tumor (Appendix A). 

We revealed that the temperature on the skull’s external surface increased by 0.24 °C after 70 mW laser irradiation and by 1.24 °C after a 100 mW laser pulse of 17 min. Importantly, this was not associated with any changes in the temperature of the cortex surface, neither after 70 mW (37.12 ± 0.11 °C) nor after 100 mW (37.19 ± 0.11 °C) laser irradiation for 17 min. Laser radiation of a 130 mW intensity increased the temperature by 2.18 °C on the skull and by 1.87 °C on the cortex surface. Thus, the laser intensities of 70 mW and 100 mW did not induce any significant increase in the temperature of the brain tissues and could not affect brain morphology and functioning.

In the next step, we studied the effects of different laser irradiation doses on survival rate and GBM volume. Figure 1b illustrates the laser beam locations and the algorithm of the LT course: 17 min laser irradiations with a 5 min pause between them for 61 min every day for 4 weeks. The survival rate and four-week GBM volume were studied in two groups: (1) rats with GBM without LT and (2) rats with GBM receiving the LT course with doses of 8.7 kJ/cm^2^, 12.6 kJ/cm^2^ and 16.3 kJ/cm^2^, respectively. 

The survival rate was evaluated using the Kaplan–Meier method. A comparison of survival curves was conducted using the log-rank test. According to this test, the null hypothesis is that there is no difference between the population survival curves. Further calculations were performed using the X2 test to determine the significance of the differences (Appendix A). The survival rate in the GBM + LT group compared with the GBM group was 1.8-fold higher after the 12.6 kJ/cm^2^ LT course (64% and 34%, respectively) and 1.6-fold higher after an LT course of 16.3 kJ/cm^2^ (57% and 34%). There was no difference in survival rate between the GBM rats with LT of 8.7 kJ/cm^2^ (n = 30) and with no NIR laser applied.

The four-week-old GBMs showed the maximal tumor volumes in all animal groups (Appendix A). There were similar LT effects on GBM suppression after the LT courses with 12.6 kJ/cm^2^ and 16.3 kJ/cm^2^. Indeed, in the GBM animals receiving 12.6 kJ/cm^2^ and 16.3 kJ/cm^2^ LT, GBM volume decreased by 1.5-fold (152 ± 12 mm^3^ and 225 ± 9 mm^3^) and by 1.4-fold (159 ± 15 mm^3^ and 225 ± 7 mm^3^) compared with the GBM group. The lower LT dose of 8.7 kJ/cm^2^ did not change GBM size (233 ± 25 mm^3^ and 225 ± 9 mm^3^). 

Based on the data presented in Figure 1c and in Appendix A, the highest-efficiency LT course in terms of survival rate and reduction of GBM size without a critical increase in the temperature of the skull and the brain cortex was 12.6 kJ/cm^2^ (100 mW laser power). Therefore, this dose was chosen as the optimal one in our further experiments. 

To confirm our conclusion, we performed additional MRI analysis of the GBM volume in the independent study presented in Figure 1e,d and in Appendix A. New findings confirmed that an LT course of 12.6 kJ/cm^2^ effectively suppressed GBM growth. Therefore, GBM volume after the fourth week of the tumor receiving LT at 12.6 kJ/cm^2^ was 2.2-fold less than in rats without LT (225 ± 77 mm^3^ and 99 ± 48 mm^3^, respectively).

Using our original model of fluorescent GBM, we revealed in five rats that the 12.6 kJ/cm^2^ LT course was associated with changes in the character of tumor progression. Indeed, Figure 1f exhibits a highly diffusive GBM shape (4 weeks old) in the group without LT and likely benign tumors in the groups receiving LT courses.

### 2.2. Cellular Mechanisms of LT-Induced Suppression of GBM Growth

Next, we studied the cellular mechanisms of GBM cell death induced by LT. In our previous work using different cell lines, we showed that LT induces the production of reactive oxygen species (ROS), which is one of the mechanisms of cell death [54]. Therefore, we analyzed intracellular ROS production in response to LT-induced ^1^O_2_ generation (a single 17 min 100 mW laser irradiation) in neuronal cell models including the blood–brain barrier (BBB) model and BBB-C6 glioma cells using dihydroethidium (DHE) fluorescent single-cell imaging. The data presented in Figure 2a display the confocal imaging of oxidized DHE fluorescence intensity (DHE-FI) in cell culture models before and after LT. This qualitative analysis found that LT increased DHE-FI in both the BBB and the BBB-C6 glioma models.

Figure 2b illustrates the quantitative analysis of DHE-FI changes before and after LT in cell lines. The results unambiguously demonstrate that DHE-FI was 2.7-fold higher in BBB + LT compared with BBB (1437.7 ± 63.4 and 514.4 ± 56.4 a.u., respectively) and 2.0-fold higher in BBB-C6 + LT versus BBB-C6 (2790.5 ± 157.1 and 1329.1 ± 76.4 a.u., respectively). Thus, these data uncover a statistically significant increase in the production of intracellular ROS after LT, which was 1.9-fold higher in BBB-C6 glioma cells compared with BBB cells (2790.5 ± 157.1 a.u. and 1437.7 ± 63.4 a.u., respectively).

To evaluate the apoptosis rate in response to LT, the TUNEL method was used. Figure 2c,d shows that LT induced a statistically significant increase in apoptotic cell death. The results clearly demonstrate that TUNEL-positive cells were 2.0-fold higher in the BBB + LT model than in the BBB model without LT (20.1 ± 3.3% vs. 9.9 ± 3.2%, *p* < 0.05) and 1.5-fold higher in BBB-C6 glioma + LT than in BBB C6 glioma without LT (27.8 ± 4.2% vs. 16.1 ± 4.0%, *p* < 0.05). These data reveal that LT induction of cell apoptosis was significantly higher in BBB-C6 glioma than in the BBB model.

ROS play an important role in triggering an increase in lysosome-dependent cell death via Bax-mediated permeability of the mitochondrial membrane [62]. For a better understanding of whether LT-induced suppression of GBM growth can be related to mitochondrial pathways of apoptosis, we analyzed the expression of Bax in rats with GBM treated and not treated by LT (12.6 kJ/cm^2^). 

Our ICH data demonstrate that the expression of Bax increased 13.5 times in the GBM + LT group compared with the GBM animals (95 ± 2 and 7 ± 1% respectively) (Figure 3a(1–4),b). The Bax gene is activated in p53-apoptotic cells [63], which induces a receptor-dependent pathway of apoptosis [64]. After LT, we obtained a high expression of the P53 gene (31 ± 4%,) with a significantly lower expression of the Fas gene (2 ± 1%) (Figure 3a(5–12),b). At the same time, the products of these genes were undetectable in the GBM group (data are not presented due to P53 and Fas levels being below the method’s sensitivity).

Thus, our ex vivo and in vitro results clearly demonstrate LT-induced apoptosis in GBM cells via mitochondrial (predominant) and receptor (less represented) pathways.

When a cell dies, its components are delivered inside its lysosomes or vacuoles and are subjected to degradation in them [65]. Using the 12.6 kJ/cm^2^ LT course, we observed in in vitro (confocal and transmission electron microscopy (TEM)) and in ex vivo (immunohistochemical assay, IHC) experiments a significant decrease in the number of vacuoles as well as in the expression of proteins playing a major role in their formation. Indeed, Lc3s (MAP1-LC3a, b and c) are structural proteins in autophagosome membranes, which are widely used as biomarkers of programmed cell autophagy [66]. The IHC data (Figure 4a(1–4),b) clearly show that the expression of Lc3b was 3.6-fold lower in the GBM + LT group than in the GBM group (26 ± 5 and 95 ± 1%, respectively) Additionally, we analyzed the expression of clathrin and caveolin, playing key roles in the formation of coated vesicles [67,68]. Our findings demonstrate an essential suppression of the expression of these proteins in the GBM + LT group compared with the GBM group (23 ± 6% versus 94 ± 2%, *p* < 0.01, for clathrin, and 7 ± 2% versus 18 ± 1%, *p* < 0.01, for caveolin) (Figure 4a(5–12),b). Figure 4a(13–16),c demonstrate the in vitro data from confocal and TEM imaging of C6 glioma cells where there was a decrease in intracellular vesicles in the GBM + LT group versus GBM without LT (2 ± 1 and 26 ± 3, respectively).

GBM growth was characterized by highly proliferating tumor cells that were significantly suppressed by the LT course. Indeed, the expression of Ki67 was 1.6-fold lower in GBM + LT than in GBM without LT (Figure 3a(13–16),b).

There were no differences in the expression of any tested markers between the sham groups without versus after a 12.6 kJ/cm^2^ LT course (Figure 3a(1,2,5,6,9,10,13,14) and Figure 4a(1,2,5,6,9,10) and Appendix A). Lc3b was expressed in the cortex neurons in the sham and sham + LT groups as a positive control.

Thus, the results of this series of experiments strongly show that LT induces oxidative stress causing GBM cell apoptosis (predominantly via mitochondrial pathways involving Bax and p53) and the suppression of tumor cell proliferation associated with a decrease in vacuole formation and the expression of Lc3b as a marker of autophagy.

### 2.3. LT Causes Activation of the Drainage and Clearance of the Brain Tissues

Our findings, presented in Figure 5a–d,e, demonstrate that GBM progression (4 weeks of tumor growth) was accompanied by the development of intracranial hemorrhages (ICH) associated with perivascular edema and elevated ICP (88 ± 3 mmHg vs. 8 ± 2 mmHg) in all tested rats. With the use of a 12.6 kJ/cm^2^ LT course we found that LT significantly reduced ICP in laser-treated rats with GBM, compared with the untreated group (35 ± 2 mmHg and 88 ± 3 mmHg) (Figure 5e). We assume that this may be due to the LT-mediated activation of lymphatic drainage and clearing of the brain tissues that have been shown in our previous publications [44,46,47,48,51,52]. Indeed, Figure 5f illustrates that a single 17 min 100 mW laser irradiation significantly increases the lymphatic removal of gold nanorods (GNRs) from the cortex and draining of GNRs into the deep cervical lymph nodes (dcLNs) in both sham rats and rats with GBM. The evaluation of cerebral blood flow (CBF) by optical coherence tomography (OCT) did not reveal any changes in blood macro- and/or microcirculation after a single 100 mW laser irradiation for 17 min, which excludes immediate effects of the 1267 nm laser on cerebral hemodynamics (Appendix A). There were no morphological changes in the brain vessels and tissues either (Appendix A). 

Additionally, to answer the question of whether LT directly affects lymphatic drainage and clearance of the brain tissues independent of GBT progression, we analyzed LT’s effects on the diffusion and lymphatic clearance of FITC–dextran 70 kDa (FITCD). Figure 5g–l demonstrates confocal imaging of the distribution and removal from the brain of FITCD injected into the right lateral ventricle via an implanted catheter to avoid anesthesia effects on lymphatic clearance [69].

Our results clearly demonstrate that intraventricular injection of FITCD was accompanied by the distribution of a tracer in the brain of rats in all the tested groups. Figure 5g,h,j,k shows that the spreading of FITCD was higher in the ventral than in the dorsal aspect of the brain 3 h after its injection. These data reflect FITCD distribution from the ventricle to the basal MLVs, playing an important role in brain drainage and clearance [70]. dcLNs are the first anatomical station of the CSF exit, with dissolved unnecessary substances from the brain [45,47,51,52,53]. Therefore, the more intensively a substance is released from the brain, the higher its level in dcLNs. Figure 5i,l illustrates that the fluorescent signal from FITCD in the dcLNs was higher in the treated group vs. the untreated group, suggesting a stronger accumulation of tracer in rats after LT than in rats without LT. Indeed, Figure 5m–p demonstrates a quantitative analysis showing that the fluorescent signal from FITCD was significantly higher in dcLNs and lower in the brain in rats treated with LT than in untreated animals (dcLN: 0.134 ± 0.06 p.u. vs. 0.028 ± 0.009 p.u.; the brain: 0.083 ± 0.026 p.u. vs. 0.321 ± 0.119 p.u.). 

These results suggest that the progression of GBM is accompanied by ICH associated with perivascular edema; this can be a reason for an ICP rise, which we observed in all rats after 4 weeks of GBM growth. A 12.6 kJ/cm^2^ LT course significantly attenuates these consequences of tumor progression via LT-mediated activation of lymphatic drainage and clearance of the brain tissues, which we demonstrated in our previous work [45,47,48,49,52,53] and confirmed here in experiments with the monitoring of lymphatic clearance of tracers from the brain into the dcLNs in sham rats and rats with GBT (GNR), as well as in healthy rats (FITCD). 

To study the importance of the lymphatic system in GBM progression, we analyzed the effects of a blockade of the lymphatic pathways of brain fluid outflow via the cervical lymphatic vessels and the cribriform plate on the survival rate of rats with GBM (Appendix A). Our results showed a decline in the ability of animals to resist GBM progression, which was accompanied by a significant decrease in their survival rate (21% vs. 34%) (Figure 1d and Appendix A).

## 3. Discussion

In this study, we demonstrated that 1267 nm LT causes oxidative stress. In our earlier study, we showed, using naphthacene, that 1268 nm irradiation bleaches naphthacene in a dose-dependent manner in the presence of air or oxygen in tetrachloride solution, which is solid evidence of ^1^O_2_ generation with a 1268 nm laser [54]. Naphthacene does not absorb 1267, neither can it be oxidized by triplet oxygen. Besides this evidence, in our other study we detected, using Si-DMA and a Singlet Oxygen Sensor Green (with high ^1^O_2_ selectivity) increased production of ^1^O_2_ in neurons and astrocytes after 1268 nm irradiation [71]. In this paper, we used DHE to measure the bulk production of ROS, including the superoxide anion, which is a first byproduct of ^1^O_2_ [72]. Thus, our data revealed that LT’s therapeutic effects on GBM are largely realized through oxidative stress. At the same time, taking into account the absence of reliable markers of singlet oxygen in a living organism, in particular in rodent brain, the question of the contributions of ROS and singlet oxygen to LT-mediated oxidative stress remains open and requires more detailed study. 

Our findings reveal that LT-induced oxidative stress triggers GBM cell apoptosis, predominantly via a mitochondrial pathway (Bax and p53) associated with vacuole formation and the suppression of tumor cell proliferation that critically decreases GBM volume and increases survival rate in rats. We assume that LT-mediated apoptotic events via an intrinsic cell pathway can be important for targeting chemotherapy in GBM, which has been discussed as the most preferred method for GBM cell death [73,74]. Vacuole-inducing compounds can disrupt endolysosomal trafficking and stimulate production of exosomes by the GBM cells [75]. The LT-mediated modulation of vacuole formation in GBM might be an important biomarker in itself for the effectiveness of GBM therapy, as well as providing changes in the value of exosomes, not only as diagnostic and prognostic markers of GBM progression [76,77].

We carefully investigated the role of temperature in the therapeutic effects of LT on GBM. Our data show that laser power intensities of 70 mW and 100 mW did not induce any significant increase in the temperature of brain tissues and could not affect brain morphology and functioning. Thus, we show that the therapeutic effects of LT on GBM are temperature-independent. In our previous study, we also demonstrated that with a power level significantly higher than used in our manuscript temperature rise did not exceed 4 C^0^ [54]. Additionally, we revealed temperature dependence on induced laser power, which clearly excludes temperature as a main therapeutic effect [78].

Currently, immunotherapy for brain tumors is being discussed as the most promising strategy in combating GBM [79,80]. Our results demonstrate that an LT course significantly attenuates ICP, the main reason for ICH and vasogenic edema accompanying GBM progression [81,82]. In our recent research, we have shown that a 1267 nm laser stimulates the drainage and clearing functions of MLVs [45,47,48,49,52,53], including removal of blood from the brain [83]. Based on these facts, we hypothesize an important role for lymphatic mechanisms in the LT-mediated increase in the survival rate of rats with brain tumor from 34% to 64%. We assume that LT also reduces ICH in rats with GBM via LT-mediated stimulation of the lymphatic drainage of brain fluids. Our findings demonstrate that LT increases clearance of GNRs from the brains of both sham and GBM rats, suggesting LT-related activation of the lymphatic clearing function. Additionally, we show LT stimulation of lymphatic removal of FITCD from the brains of healthy rats, excluding GBM-related influence on the LT-mediated increase in lymphatic drainage and clearing of the brain tissues. Our results regarding the lymphatic blocking of brain fluid outflow exhibit a significant decrease in survival rate (from 34% to 21%) in rats with GBM, confirming a crucial role of the lymphatic system in mechanisms of resistance to brain tumor growth. Song et al. discovered that vascular endothelial growth factor C (VEGF-C), which is expressed in the lymphatic endothelium, including in MLVs, promotes enhanced priming of CD8 T cells in the dcLNs, migration of CD8 T cells into GBM, and long-lasting anti-tumor memory response in rats [42]. They demonstrated that VEGF-C-driven lymphatic drainage enables immune surveillance of GBM. 

Taking into account the important role of the lymphatic system in resistance to GBM, we assume that augmentation of lymphatic function might be a promising therapeutic target for preventing or delaying brain tumor growth. Among several possible methods to induce an effective anti-tumor immune response, LT might have demonstrated strong potential.

## 4. Materials and Methods

### 4.1. Subjects and Groups

Pathogen-free male Wistar rats (200–250 g, 2 months old) were used in all experiments and were obtained from the National Laboratory Animal Resource Centre (Pushchino, Moscow, Russia) or Charles River Laboratories, Inc. (Wilmington, MA, USA). The animals were housed under standard laboratory conditions, with access to food and water ad libitum. All experimental procedures were performed in accordance with the “Guide for the Care and Use of Laboratory Animals”, Directive 2010/63/EU on the Protection of Animals Used for Scientific Purposes, and guidelines from the Ministry of Science and Higher Education of the Russian Federation (No. 742 from 13.11.1984), which have been approved by the Bioethics Commission of the Saratov State University (Protocol No. 7) and the Institutional Animal Care and Use Committee of the University of New Mexico, USA (19–200767-HSC200247). Experiments were divided into 4 groups: (1) sham rats without the LT course; (2) sham rats + the LT course; (3) rats with GBM without the LT course; and (4) rats with GBM + the LT course. Appendix A illustrates schematically the design of the experiments and the number of rats in the different groups. The calculation of the rat sample size was based on recommendations from [84]. However, due to general requirement to minimize the number of animals sacrificed, we had in total 468 rats (Appendix A). Power analysis was used to identify the minimum number of animals needed to avoid a Type II Error. The assumptions in our power analysis were a fixed level of 0.05 with a planned two-tailed pair-wise comparison, a standard deviation of 15 percent and a 40 percent difference between the groups as substantively important. These calculations determined that n = 5 per data point was required to detect a statistically significant difference.

All experiments with animals were performed according to ARRIVE guidelines.

### 4.2. Implantation of C6 and Fluorescent Glioma Cell Lines [85]

There are several reasons for a C6 glioma model choice. The vast majority of publications devoted to study of the phototherapy of GBM have been made using the C6 glioma line [86,87,88]. The C6 cancer cell line is a rat glioma cell line, which can simultaneously simulate the high growth rate, high vascularization, and highly infiltrative character of glioblastoma multiforme. Therefore, the C6 glioma cell line is widely used for the study of GBM therapy and its mechanisms of development. Most C6 glioma research has been focused on testing a wide diversity of agents for their tumoricidal activity. The C6 cell line is considered to be a safe and popular glioma model in the literature, providing a good simulation of glioblastoma multiforme [89]. The C6 glioma model was generated from an ENU-induced glioma in an outbred strain of Wistar rats. As a consequence, inoculation of C6 cells in common Wistar rat strains results in an allogenic immune response and a lack of tumor growth [90]. However, the cell lines that do grow in their syngeneic hosts after intracerebral transplantation develop into invasive cancers that have been used to investigate effects of targeted therapy and radiotherapy [91].

The rats were pre-treated using premedication with Seduxen (Gedeon Richter, Hungary) at a dose of 50 μg/mL. Afterward, the rats were deeply anesthetized with intraperitoneal Zoletil (Virbac, France) at a dose of 100 μg/kg, moved into a stereotaxic head holder and immobilized on the stereotactic system (Narishige, Japan) by fixation of the head. The scalp of the anesthetized rat was shaved and scrubbed with betadine 3 times followed by an alcohol rinse. Hair was removed at the site of the planned operation and a cut was made in the area of the planned injection. An incision was made over the sagittal crest from the bregma to the lambdoidal suture and the periosteal membrane was removed. A small dental drill was used to create a burr hole through the bone without tearing the dura matter in the exposed cranium, 0.5 mm anterior and 3 mm lateral to the bregma. 

The C6 rat glioma cell line was obtained from the Russian Cell Culture Collection of Vertebrates, Institute of Cytology, Russian Academy of Sciences (St. Petersburg, Russia), or the American Type Culture Collection (Manassas, VA, USA). A transfected C6—TurboFP635 cell line was used for the study of the growth of fluorescent GBM [82]. C6 cells were cultured in a Dulbecco’s Modified Eagle Medium (DMEM) growth medium (Paneco, Russia) containing 2.5% embryonic veal serum (Biosera), 4 mm glutamine (Paneco, Russia), penicillin (50 IU/mL) and streptomycin (50 mg/mL) (Paneco, Russia). Rat C6 glioma cells were transfected with turboFP635-C DNA plasmids using the method of liposomal transfection followed by selection using geneticin (G418 antibiotic, neomycin analogue). The resulting cell line, C6-TurboFP635, has stable cultural and morphological characteristics.

The glioma cells (5 × 105 cells per rat) were injected at a depth of 4.5 mm from the brain surface into the caudate putamen area with a Hamilton microsyringe in a volume of 15 μL at a rate of 1 μL/min. Physiological saline (15 μL, Sigma-Aldrich, St Louis, MO, USA) was injected in the same region of the brain in the sham groups. Thereafter, the burr hole was sealed with sterile bone wax and tissue glue and the wound was sutured closed with 3-0 absorbable suture material. After the implantation of glioma cells, the wound was closed and treated with 2% brilliant green solution. The rats were removed from the stereotaxic head holder, given 0.01 mg/kg buprenorphine, s.c., and 50 K bicillin, i.m., returned to their cages after recovery in a temperature-controlled recovery cage and moved back to the animal facility after recovery. The animal was placed in a clean cage. GBM growth and tumor volume were monitored by MRI 7 days post tumor cell implantation using the Clin scan 7T tomograph (Bruker, Mannheim, Germany). The growth of fluorescent GBM C6-TurboFP635 in the sham group and in rats that received the LT course was assessed by confocal microscopy using a Leica SP5 confocal laser scanning microscope (Leica, Munich, Germany) and 3D images were obtained using the LAS x software platform for the LSX Life Science microscope. Survival was expressed as % and calculated as the ratio of the number of dead rats divided by the total number of animals at the beginning of the experiment, expressed as 100%. Comparison of the two survival curves was conducted using a statistical hypothesis test called the log-rank test. This is used to test the null hypothesis that there is no difference between the population’s survival curves. Further calculations were performed using the X2 test to determine the significance of the differences. With the X2 test, the survival differences between the groups were significant (*p* = 0.00001437) [92].

### 4.3. Laser Radiation Scheme and Dose Calculation

#### 4.3.1. LT Effects: In Vivo Study

A fiber Bragg grating wavelength-locked high-power laser diode (LD-1267-FBG-350, Innolume, Dortmund, Germany, laser driver: THORLABS CLD1015) emitting at 1267 nm was used as a source of irradiation. The laser diode was pigtailed with a single-mode distal fiber ended by the collimation optics to provide a 5 mm beam diameter at the specimen (Appendix A). The rats had recovered 7 days after glioma cell transplantation. Thereafter, the heads of rats with MRI-identified glioblastoma were shaved and the scalps were removed, then they were fixed in a stereotaxic frame and irradiated in the area of the glioma cell injection with a near-infrared laser for 4 weeks every day, under inhalation anesthesia (1% isoflurane at 1 L/min N_2_O/O_2_—70/30 ratio).

Transmission analysis of the 1267 nm laser irradiation passing through a freshly prepared rat skull sample revealed a scattering effect giving a 1.2-fold wider laser beam of 6 mm, and only 35% of the initial laser energy reached the top layer of the cortex (see the measurement setup scheme in Appendix A). The laser doses were calculated as follows:D = 0.35 ∗ P/(1.2 ∗ S) ∗ T(1)
where D is the irradiation dose; S is the laser spot area on the brain cortex (cm^2^); P is the laser irradiation power on the skull surface (W); and T is the full laser irradiation time (s). The full daily LT course, comprising 17 min laser + 5 min pause + 17 min laser + 5 min pause + 17 min laser, with 70, 100, and 130 mW laser power intensities applied, gave 313, 450 and 580 J/cm^2^ daily doses for each animal, irradiated for 51 min per day in total. The whole LT course, where animals were irradiated every day for 4 weeks, gave 8.7, 12.6, and 16.3 kJ/cm^2^ doses.

#### 4.3.2. The LT Effects: In Vitro Study

Appendix A schematically illustrates the irradiation of C6 glioma cell cultures in a 6-well plate with a 1267 nm laser accomplished with a single-mode fiber light guide (LD-1267-FBG-350, Innolume, Dortmund, Germany, laser driver: THORLABS CLD1015) and a collimator. The collimator was placed 50 mm below the well plate and a parallel laser beam (5 mm diameter) was directed vertically upward through the bottom of the 6-well plate to the C6 glioma cell culture (Appendix A). Each well was irradiated individually.

For in vitro studies, the laser output power was set to 30 mW to ensure the same laser radiation intensity over the cells as for in vivo experiments. According to Equation (1), 100 mW of laser radiation power applied to a rat’s skull corresponds to an intensity of 100 ∗ 0.35/(1.2 ∗ 0.2) = 145 mW/cm^2^ at the brain surface. Thus, to ensure the same intensity at cells in the lunule irradiated with a 5 mm diameter collimated beam and assuming 5% attenuation because of the reflection and absorption of laser radiation passing through the bottom of the lunule, the reduced laser radiation power was set to 145 mW ∗ 0.2 cm^2^/0.95 = 30 mW, where 0.95 is the transmission coefficient of the plastic lunule. After the cells were irradiated following the daily LT course protocol comprising 17 min LR + 5 min pause + 17 min LR + 5 min pause + 17 min LR, the total irradiation time was 51 min and the dose was 450 J/cm^2^, which corresponds to the effect of 100 mW laser radiation power applied in vivo through the skull.

During both in vivo and in vitro experiments, laser radiation power was measured with a power meter (1815-C, with thermopile sensor head 818P-070-20, Newport Inc., Irvine, CA, USA).

### 4.4. Measure the Thermal Impact of LT

A type-A-K3 thermocouple (Ellab, Hillerød, Denmark) was used to measure skull temperature. The thermocouple was placed subcutaneously 2 mm lateral to the bregma in the irradiated zone. A burr hole was drilled under inhalation anesthesia (1% isoflurane at 1 L/min N_2_O/O_2_—70:30). To measure brain surface temperature under 1267 nm laser irradiation, the medial part of the left temporal muscle was detached from the skull bone, a small burr hole was drilled into the temporal bone, and a flexible thermocouple probe (IT-23, 0.23 mm diameter, Physitemp Instruments LLC, Clifton, NJ, USA) was introduced between the parietal bone and the brain into the epidural space. Brain surface temperature was measured before and during laser stimulation with a 5 min increment using a handheld thermometer (BAT-7001H, Physitemp Instruments LLC, Clifton, NJ, USA).

### 4.5. Measurement of ICP

For ICP monitoring, a catheter (PE-50) filled with artificial cerebrospinal fluid (BioChemazone, Ontario, Canada) was inserted through the atlanto-occipital membrane into the cisterna magna and glued (cyanoacrylate glue) in place. The ICP catheter was connected to the pressure transducer TSD104A with a DA100C amplifier (Biopac Systems, Inc., Goleta, CA, USA). ICP was continuously recorded throughout the experiment using a Biopac MP160 data acquisition system and AcqKnowledge software 5.0.5. (Biopac Systems, Inc., Goleta, CA, USA).

### 4.6. Magnetic Resonance Imaging (MRI) of GBMs

Tumor volume assessment was performed on the 1st and 4th weeks after GBM implantation using a 7 Tesla Bruker BioSpec 70/30 USR dedicated research MRI scanner and Paravision 6.0 data acquisition software (Bruker Biospin; Billerica, MA, USA). To obtain a good signal-to-noise ratio, a 72 mm small-bore linear RF volume coil with an actively decoupled brain surface coil (40-cm bore; a 660 mT/m, rise time within 120 μs) was used for excitation and signal detection, respectively. Rats were anesthetized with 2% isoflurane at 1 L/min N_2_O/O_2_—70:30. The animal was placed into a stereotaxic headset and onto an animal holder with a surface coil for brain imaging. Temperature and respiration were monitored and maintained by a thermal air blower. Anatomical T2-weighted images were acquired with a fast spin-echo sequence (rapid acquisition with relaxation enhancement (RARE)) [Repetition Time (TR)/Echo Time (TE) = 5000 ms/56 ms, Field of View (FOV) = 4 cm × 4 cm, slice thickness = 1 mm, slice gap (inter-slice distance) = 1.1 mm, number of slices = 12, matrix = 256 × 256, number of averaging = 3] as previously described [93]. T1-weighted imaging used the RARE technique with 9.6 ms TE, 1000 ms TR and a RARE factor of 2, thus 4 averages, requiring 4 min 16 s. to assess tumor volumes; ROIs were drawn around regions of visible hyperenhancement in each of the slices on T2-weighted and corresponding T1-weighted images using NIH ImageJ and calculated using MATLAB software (version 2018b, MathWorks, Inc., Natick, MA, USA) [93]. To avoid prejudice in the collecting and analyzing of data, each animal to be studied received a unique code and was randomly allocated to experimental conditions, which were predetermined for each rat. All studies were done in a blind manner. The MRI operator did not know the group of the animal. Data analysis was done by analysts blind to the treatment of the rats. After completion of data analysis for each animal, the data were un-coded, grouped, statistically analyzed using GraphPad Prizm v.6.0, and compared.

### 4.7. Detection of Vacuoles In Vitro and In Ex Vivo Experiments

C6 rat glioma cells, at a concentration of 106 cells/mL, were grown on a 24 × 24 mm cover-slip placed in a 6-well tablet. Cells were divided into 2 groups: the intact cell group and the LT cell group. At the 70% confluent monolayer, when the cells were in an exponential growth phase, the old culture medium was removed and fresh DMEM medium added. After 24 h of incubation, the LT group cells were irradiated with a 1267 nm laser according to the scheme 17 min of irradiation—5 min pause—17 min of irradiation—5 min pause—17 min of irradiation (the total dose was 12.6 kJ/cm^2^) (see “Laser radiation scheme and dose calculation”). 

Confocal imaging of vacuoles. After 24 h of incubation, the growth medium was removed and 30 μM propidium iodide (PI) in DMEM medium was added to the wells. After 10 min of incubation, the PI solution was removed and 1 μM Acridine Orange (AO) (“Thermo Fisher Scientific”, Waltham, MA, USA) was added to the wells. The plates were incubated for 10 min and then washed three times with Hank’s balanced salt solution (HBSS). PI staining for the visualization of cell nuclei and AO staining for the differential staining of nucleic acids and lysosomes were carried out in order to determine the apoptotic cells according to standard protocols (Hoechst-33342-imaging-protocol, “Thermo Fisher Scientific”, Waltham, MA, USA) and the protocol presented in [94], respectively. 

Microscopic observations were conducted using a Leica DM6000B microscope (“Leica Microsystems”, Munich, Germany) with ×100, ×200 and ×400 magnification, including the use of immersion oil lenses; for confocal microscopy, a TCS SP5 confocal laser microscope (“Leica Microsystems”, Germany) was used. The signals were recorded using a λ 405 nm diode laser and an argon multilinear laser with excitation bands of λ 488 nm and λ 514 nm. Emission was recorded using the λ 460–480 nm, λ 496–506 nm and λ 525–560 nm channels. Images were obtained employing oil immersion lenses (×100). Scanning was carried out with resolutions of 1024 × 1024 and 2048 × 2048 and digital magnifications of ×1000 and ×3000. The obtained images were processed at the workstation of a confocal microscope Leica TCS SP5 using LAS X v.3.7.4 software (“Leica Microsystems”, Germany).

Transmission electron microscopy (TEM) assay of vacuoles. Microscopic observations were conducted using a Leica DM6000B microscope (“Leica Microsystems”, Germany) with ×100, ×200 and ×400 magnification, including the use of immersion oil lenses; for confocal microscopy, a TCS SP5 confocal laser microscope (“Leica Microsystems”, Germany) was used. The signals were recorded using a λ 405 nm diode laser and an argon multilinear laser with excitation bands of λ 488 nm and λ 514 nm. Emission was recorded using the λ 460–480 nm, λ 496–506 nm and λ 525–560 nm channels. Images were obtained employing oil immersion lenses (×100). Scanning was carried out with resolutions of 1024 × 1024 and 2048 × 2048 and digital magnifications of ×1000 and ×3000. The obtained images were processed at the workstation of a confocal microscope, Leica TCS SP5 (“Leica Microsystems”, Germany).

The C6 rat glioma cells were prepared for TEM as described in [95]. Electron microscopic visualization of the ultrastructure was performed on a Libra 120 electron microscope (Carl Zeiss, Aalen, Germany) at a voltage of 120 kV, with magnification range of 2500–10,000×. To perform the electron microscopic examination, the sample preparation procedure was carried out as follows: 1 mL of cell suspension was precipitated by centrifugation at 1000 rpm for 5 min and fixed with 0.1% glutaraldehyde (GA) in a 0.02% Versin solution for 1 h at room temperature. Further fixation was carried out in 3% GA prepared on the same buffer for 12 h + 4 °C. The next steps were washing of the GA with a buffer three times with an exposure of 10–15 min then post-fixation of 1% OsO4 for 3–4 h at room temperature. Dehydration was carried out at the following ethanol concentrations: 30%, rinsing several times; 50%, with exposure twice for 15 min; 70%, with exposure twice for 15 min; 90%, with exposure twice for 15 min; 100%, with exposure twice for 15 min; in acetone three times for 20 min; and in propylene oxide twice for 30 min. The resin was prepared in advance: Epon 812–15 mL, DDSA-6 mL, MNA-9 mL, and DMP 30–0.6 mL. Impregnation with resin and propylene oxide (OP) was then carried out for 12 h at the following ratios: 2:1 (OP-resin)—day, 1:1 (OP-resin)—night, 1:2 (OP-resin)—day, pure resin—night. After that, the samples were placed in molds for polymerization, which was carried out for 24 h at 37 °C, 24 h at 45 °C, and 24 h at 57 °C. Using the Reichert-Jung UltraCut E Ultramicrotome, sections of the sample with a thickness of 50–70 nm were made, placed on electron microscopy meshes (50–100 mesh, with and without a formvar substrate) and dried in air. The images obtained were processed at the workstation of a Libra 120 electron microscope using WinTem/iTem software 4.1. (Carl Zeiss, Aalen, Germany).

### 4.8. Immunohistochemical (IHC) Assay

Rats were euthanized with an intraperitoneal injection of a lethal dose of ketamine and xylazine and intracardially perfused with 0.1 M of PBS for 5 min. Afterward, their brains were removed and fixed in 10% buffered formalin, with wiring material in alcohols, pouring into paraffin. Paraffin sections were stained with hematoxylin and eosin, and IHC studies were performed using the REVEAL Polyvalent HRP-DAB Detection System. Monoclonal antibody Abcam (USA): Bcl-2/bax (MAB8272), LC3b (ab192890), Ki67 (clone SP6, ab16667), clathrin (clone EPR24231-72, ab271185), caveolin (clone E249, ab32577), Fas (ab216636), p53 (ab131442), and LC3b (ab192890) were used at a dilution of 1:100 to the antibody. When staining with ICH markers, positive and negative controls were used to exclude false-negative and false-positive results, to create standardization of the staining conditions and increase the objectivity of the results. The percentage of positively expressing cells in 10 fields of view for each sample and the intensity of immunohistochemical reactions (weak, moderate and pronounced) were calculated. All studies were performed using a MicroVisor of medical transmitted light μVizo-103 (LOMO, St Petersburg, Russia) with a magnification of 774.

### 4.9. Apoptosis Assessment

The number of apoptotic cells was estimated with the TUNEL method using the “17-141 TUNEL Apoptosis Detection Kit” (Abcam, Cambridge, UK) in accordance with the standard protocol of the manufacturer. Incubation was carried out in a reaction mixture with TdT and nucleotides labeled with avidin-FITC. Propidium iodide with RNase from another kit was added for 15 min at room temperature to stain cell nuclei. An Olympus FV10i-W confocal laser scanning microscope with aqueous immersion was used (Olympus, Tokyo, Japan). TUNEL-positive cells (cells in apoptosis) were counted, which were expressed as a percentage of the total number of cells in the field of view for at least 10 fields of view.

### 4.10. Endothelial Neuronal Cells In Vitro BBB Model

The experiments were performed on an in vitro model of the blood–brain barrier (BBB) consisting of endothelial and astroglial neural cells and macrophages. Endothelial cells were obtained from brain microvessels (BMV) of Wistar rat brains (P10). Isolation and preparation of a primary BMEC culture were performed according to the protocol of Liu et al. [96]. The resulting BMVs were phenotyped with monoclonal antibodies to the endothelial marker (ZO1) using a standard immunohistochemistry protocol using primary anti-ZO1 antibodies (Santa Cruz Biotechnology, Inc., Dallas, TX, USA, sc-8147) and secondary antibodies labeled with Alexa Fluor 488 (Abcam, Cambridge, UK ab150117) followed by the use of an Olympus FV10i-W confocal laser scanning microscope (Olympus, Japan). Astroglial and neural cells for the in vitro BBB model were derived from embryonic neurospheres, as described in [97].

### 4.11. Analysis of the ROS Production

Assessment of the level of intracellular ROS was performed fluorometrically with the dihydroethidium (DHE) approach [98]. This assay is used to detect active singlet oxygen radicals in cells in the presence of other ROS. A 30 mM DHE solution was prepared by dissolving 10 mg of DHE (95%, Sigma Aldrich) in 1 mL of DMSO [99]. Cells were seeded in glass-bottomed Petri dishes to form BBB and mixed BBB cell cultures and C6 glioma. After 24 h, DHE was added to the cocultures at a quantity of 1 μL per 1 mL of medium and incubated for 30 min. Control plates were washed with nutrient medium after 30 min and imaged with a confocal microscope (λex = 540 ± 25 nm; λem = 600 ± 40 nm) for 10 min (Time Laps). The experimental plates were washed with nutrient medium and irradiated with a 1267 nm laser as described in Section 4.3.2. The effects of LT in the in vitro study were examined with a confocal microscope under the same conditions. Confocal microscopy was performed using a fully automated confocal laser scanning microscope with water immersion, the Olympus FV10i-W (Olympus, Japan). Fluorescence level and quantitative data on fluorescence intensity were calculated using FV10-ASW 4.0 microscopy software (Olympus, Japan). Results are expressed as mean ± standard deviation. Statistical analysis was performed using non-parametric tests (Wilcoxon Matched Pairs, Mann–Whitney U Tests), and a statistically significant difference was found between the groups (*p* < 0.01).

### 4.12. Histological Analysis of the Brain Tissues and Vessels

This was done on 15 rats with GBM (4 weeks after glioma cell injection) and on healthy rats before, during and one hour after LT (n = 5 in each group). All rats were euthanized with an intraperitoneal injection of a lethal dose of ketamine and xylazine. Afterwards, their brains were removed and fixed in 10% buffered paraformaldehyde. The paraformaldehyde-fixed specimens were embedded in paraffin, sectioned (4 μm) and stained with hematoxylin and eosin. The histological sections were evaluated by light microscopy using the digital image analysis system MicroVisor medical μVizo-103(LOMO, St Petersburg, Russia).

### 4.13. Measurement of Cerebral Blood Flow (CBF)

To assess the effects of one-time LT exposure 12.6 kJ/cm^2^ on changes in cerebral circulation in the macro- (in the Sagittal sinus) and microvasculature, we used a commercial swept-source Doppler optical coherence tomography (DOCT) system, OCS1300SS (Thorlabs Inc., Newton, NJ, USA), operating at a 1325 nm central wavelength and a 100 nm bandwidth. The transverse and axial resolutions of the DOCT system were 25 and 12 μm (on the air), respectively. The A-scan rate was equal to 16 kHz, which allowed us to measure absolute velocities up to ∼5.5 mm/s [100,101]. A 30 min OCT recording of CBF (a.u.) was performed on the same healthy rats (n = 10) before, during and after one-time LT exposure 12.6 kJ/cm^2^ and expressed as an average of the sum of the CBF values. Data analysis was performed using Origin 2021 b (OriginLab) and Fiji (open-source image-processing software, https://imagej.net/software/fiji/#downloads, accessed on 1 September 2023) [102].

### 4.14. Blocking of the Lymphatic Pathway of CSF Outflow

This was done through the cervical lymph vessels and the cribriform plate (Appendix A). The cervical lymph nodes, including the bilateral submandibular superficial and deep nodes, were identified, isolated and removed under a dissecting microscope in anesthetized (intraperitoneal Zoletil (Virbac, France), at a dose of 100 μg/kg, with premedication with Seduxen (Gedeon Richter, Hungary) at a dose of 50 μg/mL) rats with GBM. To remove the cribriform plate, we used the method described in Ref. [103]. Briefly, the skin over the frontal-nasal area was reflected to reveal the frontal and nasal bones. The nasal bone was removed to expose the nasal mucosa with the upper edge approximately at the level of a line bisecting the medial canthi. An external ethmoidectomy was performed; the nasal mucosa, olfactory nerves, and all soft tissue on the extracranial surface of the cribriform plate were scraped away with a curette, and the bone surface was sealed with the aforementioned tissue glue. At the end of each experiment, an Evans blue dye–protein complex was injected into the CSF compartment to check for possible CSF leaks.

### 4.15. In Vivo and Ex Vivo Optical Monitoring of Lymphatic Clearance of FITCD from Rat Brain 

Ten days before the experiments, a polyethylene catheter (PE-10, 0.28 mm ID × 0.61 mm OD, Scientific Commodities Inc., Lake Havasu City, AZ, USA) was implanted into the right lateral ventricle (AP—1.0 mm; ML −1.4 mm; DV—3.5 mm) according to the protocol reported by Devos et al. [104]. A small cranial burr hole was drilled through the skull using a variable-speed dental drill (with a 1 mm drill bit). An amount of 5 μL of FITCD 70 kDa (Sigma-Aldrich, Burlington, VT, USA), at a rate of 0.1 μL/min, was injected into the right lateral ventricle at 08:00 am in the group of awake healthy rats.

The ex vivo optical study of FITCD distribution was performed 3 h after the intra-ventricular injection of FITCD in rats. The imaging was performed using a homemade optical instrument based on the monochrome camera acA2040–2090 um (Basler, Ahrensburg, Germany) and a 50 mm ½.8 C-mount CCTV objective lens (Tamron, Minuma-ku, Saitama, Japan). The lens was attached to the camera with a 15 mm extension tube to ensure macro imaging within a field of view ranging from 23 to 32 mm depending on the lens focusing ring adjustment. The lens was mounted on the vertical manual translation stage (Standa, Lithuania) above a Petri dish, where samples were submerged in a buffer solution. The top surface of each sample was covered with a 25 mm × 50 mm × 0.17 mm cover glass. The slider with filter sets (49019, 49002, Chroma Technology, Bellows Falls, VT, USA) was placed just below the objective lens. Each filter set was illuminated with homemade condensers with 1W LEDs (635 nm for 49019 and 460 nm for 49002) to ensure uniform illumination over the camera field of view. LED illuminators were synchronized with the camera “fire” output.

The camera resolution was 2048 × 2048 pixels at 12-bit grayscale. Images were acquired in a dark room at a constant exposure time of 200 ms, and other settings were kept unchanged for all samples. Image acquisition and processing were performed with custom software developed using the NI Vision and LabVIEW software 6.1 (National Instruments, Eagan, MN, USA) and the Fiji open-source image-processing package [102]. Image-processing procedures were identical for each pair of images (control and laser-treated samples) for each channel to ensure an accurate comparison of the fluorescence intensity.

The raw integrated density of pixels in the FITCD channel in 500 × 500 (px) regions of interest was measured within the point of connection of the transverse sinuses and the superior sagittal sinus. Measurements were performed over 6 consecutive slices of the image stacks for each brain specimen. The first (top) slice to analyze was determined based on the width of the fluorescent area in the sagittal sinus, which had to fit in the width of a 500 × 500 px region of interest.

In the dcLNs, measurements were also performed in FITCD over 500 × 500 px regions of interest. The 500 × 500 region was positioned within the fluorescent area of each lymph node closest to the cover slip above the lymph node. In every lymph node image stack, the raw integrated density of pixels in the FITC dextran channel was measured in 3 consecutive slices of the stack.

### 4.16. Statistical Analysis

Microsoft Office Excel and SPSS 24.0 for Windows were used for statistical analysis. The Shapiro–Wilk test was used to find out whether the parameter values in the groups followed a normal distribution. If the Shapiro–Wilk test showed that the distribution of a parameter departed from normality (*p*-value < 0.05), a non-parametric Wilcoxon test and Mann–Whitney test were used for calculation of the median (Me), 25th and 75th percentiles Q(25–75), maximum and minimum. Using this method, median differences were determined at Z ≥ 1.96 at a significance level of *p* < 0.05 (with more than 95% probability). If the Shapiro–Wilk test did not show evidence of non-normality, we decided to use parametric tests. The results are presented as mean ± standard error of the mean (SEM). Differences from the initial level in the same group were evaluated with Welch’s *t*-test and ANOVA-2 (post hoc analysis with Duncan’s rank test) to determine whether differences existed among the means of three or more groups. Welch’s *t*-test consists of the division of the difference between the arithmetic means of two (control and experimental) samples by the natural estimate of the mean-square deviation of this. T ≥ 1.96 indicates a 95% probability of the mean values being different with this method. 

## 5. Conclusions

Our findings demonstrate that PS-free LT significantly increases the resistance of rats to GBM progression, providing effective suppression of GBM growth and an increase in the animals’ survival rate from 34% to 64%. The mechanisms of LT-induced inhibition of GBM progression include: (1) suppression of the proliferation of GBM cells; (2) a decrease in vacuole formation in GBM cells; and (3) inducement of apoptosis, which is accompanied by (4 and 5) reduction of ICP through LT stimulation of lymphatic drainage and clearing functions (Figure 6). 

Hence, PS-free LT may become a promising therapeutic approach in developing a breakthrough technology for non-invasive treatment of GBM in infants in whom PDT and radio- and chemotherapy is strongly limited, as well as in adult patients with a strong allergic reaction to PSs. It should also be emphasized that infants often show a superficial location of GBM in the brain [105,106], making non-invasive treatment of pediatric tumors with LT possible.

## Figures and Tables

**Figure 1 ijms-24-13696-f001:**
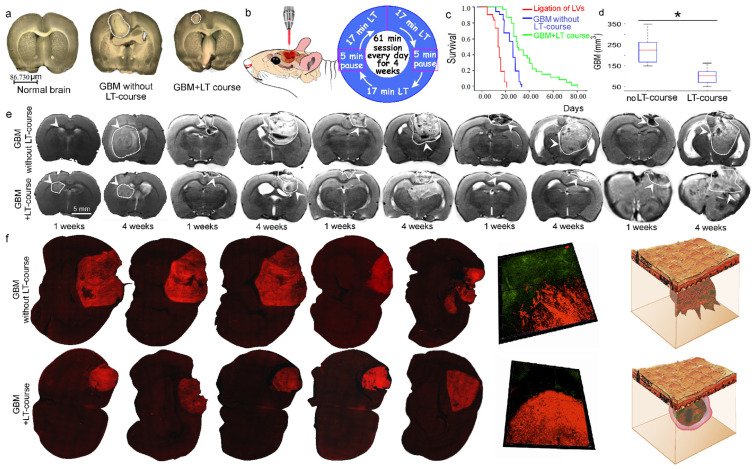
The effect of an LT course of 12.6 kJ/cm^2^ on GBM progression and survival rate: (**a**) representative 2D images of normal rat brain (left), 4 weeks of GBM growth without (middle) and after (right) the LT course, n = 5 in each group (Appendix A); (**b**) the scheme of 1267 nm irradiation applied in rats with GBM; (**c**) Kaplan–Meier overall survival plots in the tested groups without and after the LT course; the survival difference between the GBM (n = 30) and GBM + LT (n = 30) groups was significant (X2 test, *p* = 0.00001437), as well as that between the GBM group (n = 30) and the LV ligation group (n = 30) (X2 test, *p* = 0.0000000000035); (**d**) MRI data analysis of GBM volume at 4 weeks of GBM growth before and after the LT course, n = 5 in each group, * *p* < 0.01 vs. the control group (GBM without LT); (**e**) representative MRI images of GBMs (4 weeks of growth) in five rat brains before and after the LT course; (**f**) 2D and 3D images of fluorescent GBMs (4 weeks of growth) in five rat brains before and after the LT course.

**Figure 2 ijms-24-13696-f002:**
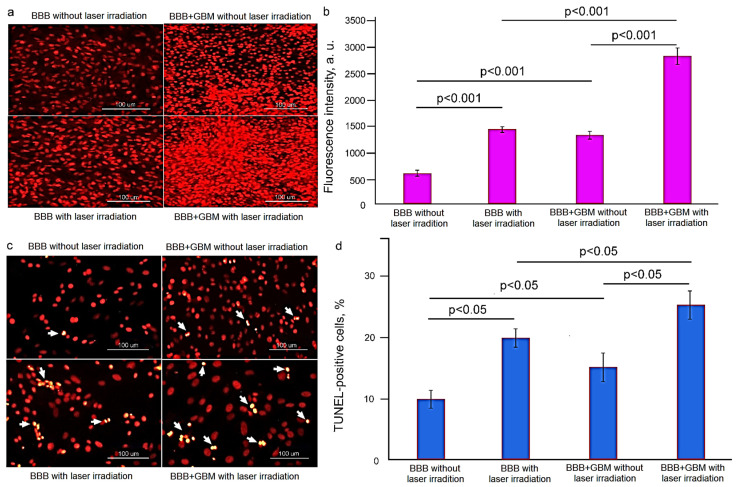
The results of LT (a single 17 min 100 mW laser irradiation) for ROS production and apoptosis: (**a**) confocal imaging of DHE-IF in the groups, including the BBB cells without LT; BBB + LT; BBB-C6 glioma cells without LT; BBB-C6 glioma cells + LT, (**b**) quantitative analysis of ROS production in the tested groups. The data are presented as mean ± SEM, n = 2 × 105 cells in each group, *p* < 0.001 between groups, Wilcoxon and Mann–Whitney U tests; (**c**) representative fluorescence images of TUNEL-positive (yellow, arrowed) cells in the groups, including the BBB cells without LT; BBB + LT; BBB-C6 glioma cells without LT; BBB-C6 glioma cells + LT; (**d**) quantitative analysis of TUNEL-positive cells (expressed as a percentage of the total number of cells in the field of view for at least 10 fields of view) in the tested groups. The data are presented as mean ± SEM, n = 2 × 105 cells in each group, *p* < 0.05–0.01 between groups, Wilcoxon and Mann–Whitney U tests.

**Figure 3 ijms-24-13696-f003:**
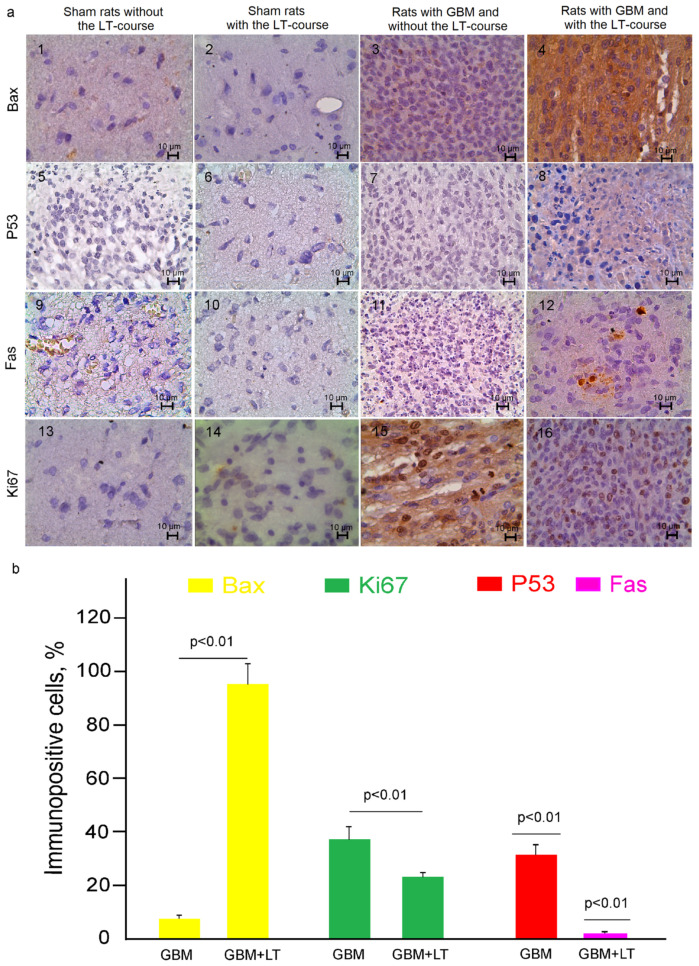
An LT course (12.6 kJ/cm^2^) affects apoptosis and proliferation in GBM cells: (**a**) (1–16)— IHC assay of the expression of tested markers in the groups, including the sham rats without the LT course, sham rats + LT, the GBM rats without the LT course, and GBM + LT, where (1–12) are the Bax, p53 and Fas markers of apoptosis, respectively and (13–16) show the Ki67 marker of the proliferation of GBM cells; (**b**) quantitative analysis of immunopositive cells (%) expressing Bax, p53, Fas and Ki67 in the tested groups; n = 20 for each group, *p* < 0.01 between groups, Welch’s *t* test).

**Figure 4 ijms-24-13696-f004:**
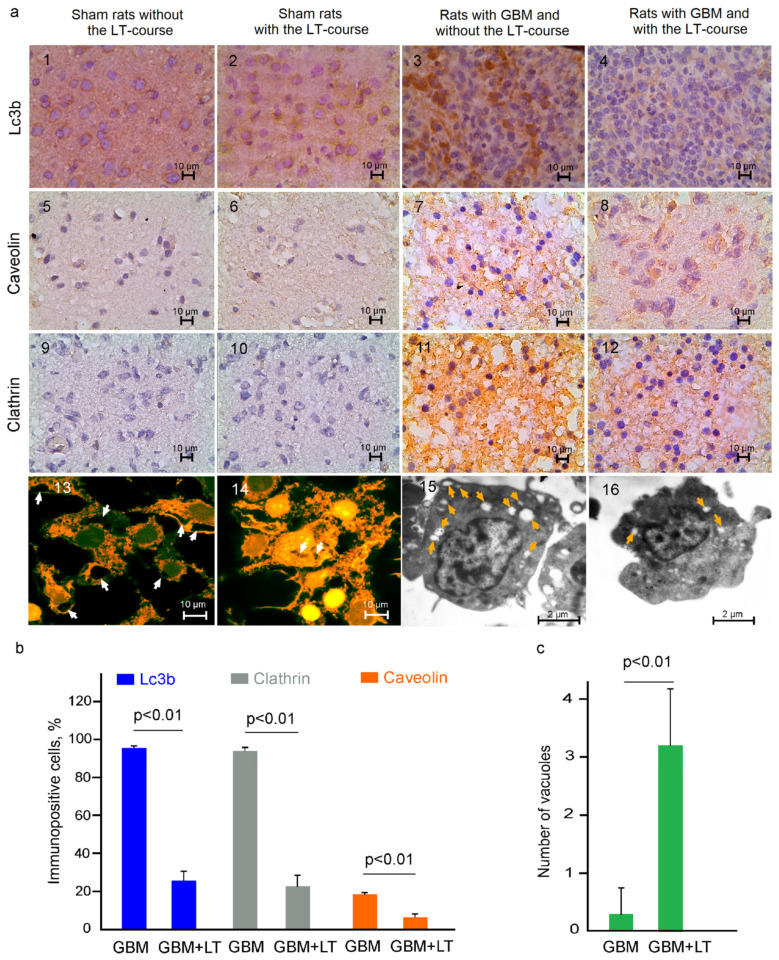
An LT course (12.6 kJ/cm^2^) affects vacuole formation in GBM cells: (**a**) (1–12)—IHC assay of the expression of tested markers in the groups, including the sham rats without the LT course, sham rats + LT, GBM without the LT course, and GBM + LT, where (1–12) show Lc3b (a marker of autophagy) and clathrin and caveolin (markers of coated vesicles), respectively; (**b**) quantitative analysis of immunopositive cells (%) expressing Lc3b, clathrin and caveolin in the tested groups; n = 20 for each group, *p* < 0.01 between groups, Welch’s *t* test); (**c**) quantitative analysis of the number of vacuoles in GBM cells without and with LT (n = 20 for each group, *p* < 0.01 between groups, Welch’s *t* test).

**Figure 5 ijms-24-13696-f005:**
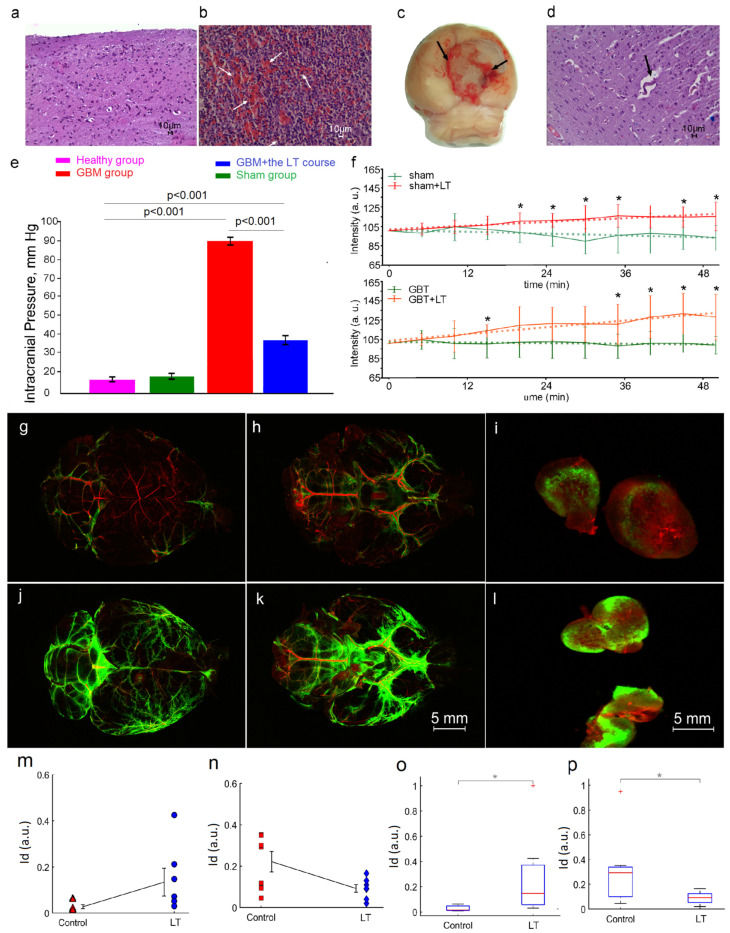
LT effects the drainage and the clearance of the brain: (**a**,**b**,**d**) histological imaging of normal brain tissues: (**a**) ICH ((**b**), arrowed) and perivascular edema ((**d**), arrowed) in rats with GBM (stained by hematoxylin and eosin). The bars (**a**,**b**,**d**) 224.6× (**d**) demonstrate a macro photo of GBM with ICH; (**c**) Macro photo of brain with hemorrhages (arrowed) around glioma; (**e**) ICP (mmHg) in the groups, including in the sham rats and in rats with GBM before and after the LT course of 12.6 kJ/cm^2^; n = 10 for each group, Wealth’s test; (**f**) OCT monitoring of GNR accumulation in the dcLNs in the groups, including in sham rats and in rats with GBM before and after LT at 12.6 kJ/cm^2^; n = 10 for each group, * *p* < 0.05 vs. rats without LT; Wealth’s test; (**g**–**l**) representative images of the distribution of FITCD in the dorsal (**g**,**j**) and ventral (**h**,**k**) aspects of the brain and the accumulation of FITCD (**i**,**l**) in dcLNs 3 h after the intraventricular injection of FITCD in the tested groups; (**m**–**p**) quantitation of the signal intensity of FITCD in the control (untreated) group and the LT group. Mean values for raw integrated density (Id, arbitrary units) in dcLNs (**m**) and in the brains (**n**) for the control group without LT and in the LT group. Data are mean ± standard error of the mean (SEM). Boxplot for raw integrated density in dcLNs (**o**) and brains (**p**) for the control group without LT and in the LT group. Boxplots represent the median (center line), interquartile range (box limits), extreme data points (whiskers), and outliers (+). Comparisons were confirmed using the two-sample Welch *t*-test and non-parametric Mann–Whitney U-test, * *p* < 0.05.

**Figure 6 ijms-24-13696-f006:**
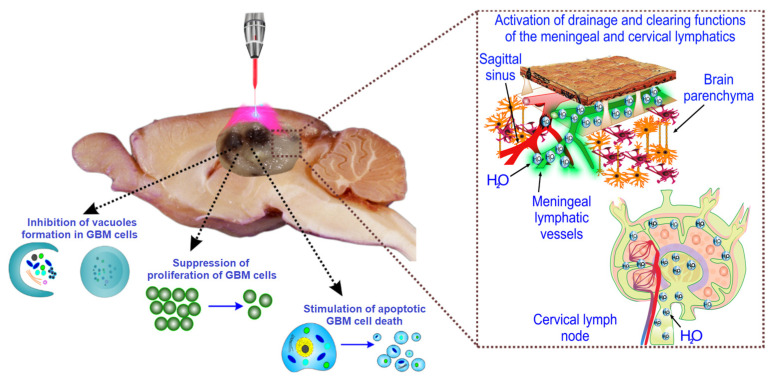
Scheme of PS-free LT therapy for GBM via LT-mediated mechanisms, such as (1) suppression of the proliferation of GBM cells, (2) a decrease in vacuole formation in GBM cells; and (3) inducement of apoptosis, which is accompanied by (4 and 5) a reduction in ICP through LT stimulation of lymphatic drainage and clearing functions.

## Data Availability

The data that support the findings of this study are available on request from the corresponding author.

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
