# Peer review of "Transcranial Photosensitizer-Free Laser Treatment of Glioblastoma in Rat Brain"

_ijms, 2023, doi:10.3390/ijms241813696_

Round 1

Reviewer 1 Report

Development of near-infrared (NIR) photosensitizer-free phototherapy would have great benefits in glioblastoma therapy.  The authors claimed that illumination by a 1267 nm laser could stimulate a direct generation of singlet oxygen from triplet oxygen without photosensitizers.  Sufficient singlet oxygen (recognized as a nonradical ROS) is lethal to cancer cells.  I’m curious about how triplet oxygen was excited, as we know that singlet oxygen emission is right at this wavelength (~1270 nm).  I noticed that the authors detected singlet oxygen using DHE as a probe.  However, DHE is usually used to detect superoxide ion which is a radical ROS, not suitable for detection of singlet oxygen.  The authors must identify the ROS induced by the NIR laser treatment to clarify the mechanism of phototherapy.  If singlet oxygen is ruled out from the major laser-induced ROS, a new understanding of therapeutic mechanism may be necessary.

Author Response

Comments: Development of near-infrared (NIR) photosensitizer-free phototherapy would have great benefits in glioblastoma therapy. The authors claimed that illumination by a 1267 nm laser could stimulate a direct generation of singlet oxygen from triplet oxygen without photosensitizers.  Sufficient singlet oxygen (recognized as a nonradical ROS) is lethal to cancer cells.  I’m curious about how triplet oxygen was excited, as we know that singlet oxygen emission is right at this wavelength (~1270 nm).  I noticed that the authors detected singlet oxygen using DHE as a probe.  However, DHE is usually used to detect superoxide ion which is a radical ROS, not suitable for detection of singlet oxygen.  The authors must identify the ROS induced by the NIR laser treatment to clarify the mechanism of phototherapy.  If singlet oxygen is ruled out from the major laser-induced ROS, a new understanding of therapeutic mechanism may be necessary.

Response: The authors express their sincere gratitude to the reviewer for constructive comments and great help in improving the quality of our article. All changes in the article are highlighted in yellow.

In our early research (Ref [54], S. Sokolovski et al. 2013, Fig. 1A) we demonstrated using naphthacene that 1268 nm irradiation in dose-dependent manner bleached the naphthacene in present of air or oxygen in tetrachloride solution which is solid evidence of singlet oxygen generation by 1268 nm laser. Naphthacene does not absorb the 1267 neither it can be oxidized by triplet oxygen. Besides this evidence in other our study (S. Sokolovski et al. Free Rad Biol and Med, 2021) we detected using Si-DMA and SOSG (have high 1O2 selectivity) an increased production of the 1O2 in neurons and astrocytes. Why in this paper we used DHE to measure bulk production of ROS including superoxide anion which is a first byproduct of the 1O2 (Halliwell, B. & Gutteridge, G. M. Free radicals in biology and medicine (Oxford University Press, 2007).

We improved Discussion and added this explanation in the manuscript:

Lines 309-317: “In this study, we demonstrate that 1267 nm LT causes an oxidative stress. In our early study, we demonstrated using naphthacene that 1268 nm irradiation in dose-dependent manner bleaches the naphthacene in present of air or oxygen in tetrachloride solution which is solid evidence of 1O2 generation by 1268 nm laser [54]. Naphthacene does not absorb the 1267 neither it can be oxidized by triplet oxygen. Besides this evidence in other our study we detected using Si-DMA and singlet oxygen sensor green (has high 1O2 selectivity) an increased production of 1O2 in neurons and astrocytes after 1268 nm irradiation [71]. In this paper we used DHE to measure bulk production of ROS including superoxide anion which is a first byproduct of the 1O2 [72]".

The authors thank the referee once again for the opportunity to improve our manuscript for its possible publication in the International Journal Molecular Science.

Reviewer 2 Report

This is an innovative scientific contribution to the community dedicatedly working towards the treatment of glioblastomas. This work showcases the effects of photosensitizer-free, laser-based treatments in in vivo rat model. The work is thorough and systematically presented and is of great interest to the general readers of IJMS.

Author Response

The authors would like to thank the referee for the positive assessment of our article and the opportunity to publish it in the Journal Molecular Science.

We send the improved version of the article in accordance with the recommendations of other 2 reviewers. All corrections in the text are highlighted in yellow.

Reviewer 3 Report

In the manuscript entitled "Transcranial photosensitiser-free laser treatment of 2 glioblastoma in rat brain" Oxana Semyachkina-Glushkovskaya and her colleagues report the effects of the transcranial application of 1267 nm laser irradiation on C6 cells growing as a tumor mass after injection into the brain of rats. Intracranial tumors of rats treated by external laser irradiation after scalp removal increased their survival rate from 34% to 64%. The authors suggest that a similar strategy could be adopted for treating high grade tumors of children growing close to the surface of the cortex

This is an interesting work complementing previous papers by some of the authors of the manuscript that used infrared laser irradiation to implement opening of the blood brain barrier and activation of lymphatic drainage system

A major problem I see is the exclusive use of C6 glioma cells. As it is also recognized by the authors, C6 cells are highly immunogenic and after intracranial transplantation in rats some animal do no develop tumors even if they were left without treatment. This makes difficult to understand if the main effect on survival of laser irradiation was given by a direct toxic effect of the induced oxidative stress on C6 cells or by an indirect effect mediated by the activation of the immune system. This is important because human glioma cells are certainly less able than C6 cells of inducing an efficient immune response. A limited trial of similar experiments on F98 transplantable high grade glioma cells will be very helpful to clarify the issue and increase the preclinical value of this study.

I the discussion they should spend a little effort to explain the differences between their proposed technique and the laser interstitial thermal therapy that is already used to treat brain tumors in humans.

Minor Points

pg 17 rows 629-30 "The experimental plates were washed with nutrient medium and irradiated with a laser" Please specify which laser at what power, at what distance from the cells etc.

Author Response

In the manuscript entitled "Transcranial photosensitiser-free laser treatment of glioblastoma in rat brain" Oxana Semyachkina-Glushkovskaya and her colleagues report the effects of the transcranial application of 1267 nm laser irradiation on C6 cells growing as a tumor mass after injection into the brain of rats. Intracranial tumors of rats treated by external laser irradiation after scalp removal increased their survival rate from 34% to 64%. The authors suggest that a similar strategy could be adopted for treating high grade tumors of children growing close to the surface of the cortex

This is an interesting work complementing previous papers by some of the authors of the manuscript that used infrared laser irradiation to implement opening of the blood brain barrier and activation of lymphatic drainage system

Comment: A major problem I see is the exclusive use of C6 glioma cells. As it is also recognized by the authors, C6 cells are highly immunogenic and after intracranial transplantation in rats some animal do no develop tumors even if they were left without treatment. This makes difficult to understand if the main effect on survival of laser irradiation was given by a direct toxic effect of the induced oxidative stress on C6 cells or by an indirect effect mediated by the activation of the immune system. This is important because human glioma cells are certainly less able than C6 cells of inducing an efficient immune response. A limited trial of similar experiments on F98 transplantable high grade glioma cells will be very helpful to clarify the issue and increase the preclinical value of this study.

Response: The authors would like to express their gratitude to the referee for useful comments and suggestions for improving our article. All changes in the article are highlighted in yellow.

Thank you so much for good advice to use other model of GBM. It is really important to have comparison of the LT-course effects on the GBM progression using different GBM models that we will do definitely in our further research. However, our findings presented in this article have been done during 3 years in different labs and in different countries, including in vivo, ex vivo and in vitro experiments as well as preparation of our original model of fluorescent GBM. Our studies include 11 series of experiments with the total number of rats 468 (Figure 6SI). Our design of experiment is based on using one model of C6 glioma in all series of experiments. There are several reasons of C6 glioma model choice. The vast majority of publications devoted to the study of phototherapy of GBM have been made using C6 glioma line (Photodiagnosis Photodyn Ther. 2014 Dec;11(4):603-12. doi: 10.1016/j.pdpdt.2014.10.007; Photodiagnosis Photodyn Ther. 2019 Jun;26:405-412. doi: 10.1016/j.pdpdt.2019.05.007; Brain Res. 2012 Jan 18;1433:153-9. doi: 10.1016/j.brainres.2011.11.048; Biomater Sci. 2018 Aug 21;6(9):2410-2425. doi: 10.1039/c8bm00604k; Lasers Surg Med. 2014 Jul;46(5):422-30. doi: 10.1002/lsm.22248; Photodiagnosis Photodyn Ther. 2008 Sep;5(3):198-209. doi: 10.1016/j.pdpdt.2008.08.001).

The C6 cancer cell line is a rat glioma cell line, which can simulate in overall the high growth rate, the high vascularization, and the highly infiltrative character of GBM multiforme. Therefore, C6 glioma cell line is widely used for the study of GBM therapy and mechanisms of development. The statistics of use “C6 glioma cell lines” in PubMed (25.08.2023) is 3599 results including recent publications in Nature Group articles (Nat Commun 9, 4777 (2018). https://doi.org/10.1038/s41467-018-07250-6; Nat Commun 8, 15144 (2017). https://doi.org/10.1038/ncomms15144; Nat Commun 9, 1991 (2018). https://doi.org/10.1038/s41467-018-04315-4).

Most of the C6 glioma research has been focused on testing a wide diversity of agents for their tumoricidal activity. C6 cell line is considered to be a safe and popular glioma model in the literature, providing a good simulation of GBM multiforme (Hippokratia. 2018; 22(3):105-112. PMID: 31641331). C6 glioma model that was generated from an ENU-induced glioma in an outbred strain of Wistar rats. As a consequence, inoculation of C6 cells in common Wistar rat strains results in an allogenic immune response and lack of tumor growth (Hum Gene Ther. 1999;10:95–101. https://doi.org/10.1007/s003359900951). However, the cell lines that do grow in their syngenic hosts after intracerebral transplantation, develop into invasive cancers that have been used to investigate effects of targeted therapy and radiotherapy (Int J Radiat Oncol Biol Phys. 2013;85:805–812. doi: 10.1016/j.ijrobp.2012.07.005). We added to the methods the explanation of our choice of the C6 glioma model to study LT-effects on the GBM progression in rats (Lines 389-402).

“There are several reasons of C6 glioma model choice. The vast majority of publications devoted to the study of phototherapy of GBM have been made using C6 glioma line [86-88]. The C6 cancer cell line is a rat glioma cell line, which can simulate in overall the high growth rate, the high vascularization, and the highly infiltrative character of glioblastoma multiforme. Therefore, C6 glioma cell line is widely used for the study of GBM therapy and mechanisms of development. Most of the C6 glioma research has been focused on testing a wide diversity of agents for their tumoricidal activity. C6 cell line is considered to be a safe and popular glioma model in the literature, providing a good simulation of glioblastoma multiforme [89]. C6 glioma model that was generated from an ENU-induced glioma in an outbred strain of Wistar rats. As a consequence, inoculation of C6 cells in common Wistar rat strains results in an allogenic immune response and lack of tumor growth [90]. However, the cell lines that do grow in their syngeneic hosts after intracerebral transplantation, develop into invasive cancers that have been used to investigate effects of targeted therapy and radiotherapy [91]”.

Comment: In the discussion they should spend a little effort to explain the differences between their proposed technique and the laser interstitial thermal therapy that is already used to treat brain tumors in humans.

Response: We added in the discussion that the therapeutic LT effects on GBM are temperature independent. Therefore, our technique differs significantly from the laser interstitial thermal therapy, i.e. they are completely different technologies.

Lines 329-336:

“We carefully investigated the role of temperature in the therapeutic LT effects on GBM. Our data show that the laser power intensities of 70 mW and 100 mW did not in-duce any significant increase in the temperature of the brain tissues and could not affect the brain morphology and functioning. Thus, we show that the therapeutic LT effects on GBM are temperature independent. In our previous study we also demonstrated that with power level significantly higher than used in our manuscript temperature was not exceed 40C [54]. Additionally, we revealed that temperature dependence on induced laser power, which clearly dismiss temperature as a main therapeutic effect [78]”.

Comment: pg 17 rows 629-30 "The experimental plates were washed with nutrient medium and irradiated with a laser" Please specify which laser at what power, at what distance from the cells etc.

Response: We added the information about laser and power in the methods (Lines 651-653).

“The experimental plates were washed with nutrient medium and irradiated with a 1267 nm laser as described in the paragraph 4.3.2. The LT effects in in vitro study examined with a confocal microscope under the same conditions”.

We also extended a description in the paragraph 4.3.2 (Lines 472-479) to clarify the geometry of laser beam delivery.

“Figure S2, B schematically illustrates the irradiation of C6 glioma cell cultures in 6-wellplate with 1267 nm laser accomplished with single-mode fibre light guide (LD-1267-FBG-350, Innolume, Dortmund, Germany, laser driver: THORLABS CLD1015) and a collimator. The collimator was placed in 50 mm below well plate and parallel laser beam (5 mm diameter) was directed vertically upward through the bottom of six-well plate to C6 glioma cells culture (FigureS2 B). Each well was irradiated individually. For in vitro studies, the laser output power was set to 30 mW to ensure the same laser radiation intensity over the cells as for in vivo experiments”.

The authors thank the reviewer once again for the opportunity to improve the quality of our article with constructive recommendations for its possible publication in the International Journal Molecular Science.

Round 2

Reviewer 1 Report

The authors have replied to my previous questions about the detection/identification of singlet oxygen.  However, concerns remain to be clarified while I read through the cited publications updated in the revised manuscript.  I don’t think it’s a wise option to detect intracellular singlet oxygen using DHE as a probe.  Figure 2a may reflect the fluctuation in the level of superoxide ion rather than singlet oxygen.  Analysis of ROS productiont using a singlet oxygen-specific probe like SOSG is required.

Author Response

Comment: The authors have replied to my previous questions about the detection/identification of singlet oxygen.  However, concerns remain to be clarified while I read through the cited publications updated in the revised manuscript.  I don’t think it’s a wise option to detect intracellular singlet oxygen using DHE as a probe.  Figure 2a may reflect the fluctuation in the level of superoxide ion rather than singlet oxygen.  Analysis of ROS productiont using a singlet oxygen-specific probe like SOSG is required.

Response: The authors thank the referee for a deep analysis of our results and for a constructive advice. Of course, in vivo determination of singlet oxygen in a living organism is a very difficult task. Since this is a short-lived molecule, there will always be a question of what we actually determine the ROS or a singlet oxygen. Even using the markers that we discussed, it is impossible to state that the LT effects in the brain are realized due to the direct generation of singlet oxygen. However, the idea of our article is not to prove the role of singlet oxygen in glioma phototherapy. Our goal is to show the possibility of phototherapy of glioma without photosensitizers (PSs), which can be alternative method for people who are allergic to PSs or in newborns for whom PSs are contraindicated. In the discussion, we speak about the oxidative stress meaning both the ROS and a singlet oxygen. We have cited our previous work, where we have shown the production of singlet oxygen using SOSG in various cell models. Now at the stage of the final review, when we have only 2 days to answer you, it is impossible to make the new studies. The results presented in our article were performed for 3 years in the different laboratories and countries. We added this limitation in the discussion (Lines 317-321, highlighted in yellow). 

The authors again express their sincere gratitude to the reviewer for the opportunity to improve the quality of our manuscript and we hope that the research presented in our article can be published in the International Journal of Molecular Science.
